# MED1 Ablation Promotes Oral Mucosal Wound Healing via JNK Signaling Pathway

**DOI:** 10.3390/ijms232113414

**Published:** 2022-11-02

**Authors:** Zhaosong Meng, Zhe Li, Shuling Guo, Danfeng Wu, Ran Wei, Jiacheng Liu, Lizhi Hu, Lei Sui

**Affiliations:** 1School of Stomatology, Tianjin Medical University, Tianjin 300014, China; 2Immunology Department, Key Laboratory of Immune Microenvironment and Disease, Ministry of Education, Tianjin Medical University, Tianjin 300014, China

**Keywords:** oral mucosa, MED1, wound healing, JNK signaling pathway, activin A

## Abstract

Mediator complex subunit 1 (MED1) is a coactivator of multiple transcription factors and plays a key role in regulating epidermal homeostasis as well as skin wound healing. It is unknown, however, whether it plays a role in healing oral mucosal wounds. In this study, we investigate MED1’s functional effects on oral mucosal wound healing and its underlying mechanism. The epithelial-specific MED1 null (Med1^epi−/−^) mice were established using the *Cre-loxP* system with C57/BL6 background. A 3 mm diameter wound was made in the cheek mucosa of the 8-week-old mice. In vivo experiments were conducted using HE staining and immunostaining with Ki67 and uPAR antibodies. The in vitro study used lentiviral transduction, scratch assays, qRT-PCR, and Western blotting to reveal the underlying mechanisms. The results showed that ablation of MED1 accelerated oral mucosal wound healing in 8-week-old mice. As a result of ablation of MED1, Activin A/Follistatin expression was altered, resulting in an activation of the JNK/c-Jun pathway. Similarly, knockdown of MED1 enhanced the proliferation and migration of keratinocytes in vitro, promoting re-epithelialization, which accelerates the healing of oral mucosal wounds. Our study reveals a novel role for MED1 in oral keratinocytes, providing a new molecular therapeutic target for accelerated wound healing.

## 1. Introduction

Oral mucosa is the major protective barrier of the oral cavity. Surgical procedures, infections, poor prostheses, trauma, and adverse stimuli can injure the oral mucosa [1,2]. Oral mucosal wounds are at risk of not healing or taking a long time to heal due to the diverse flora and environment of the mouth. A chronic wound causes pain, infection, scars, and adhesions, which lead to functional defects and reduce the quality of life [3,4,5]. Therefore, it is imperative to find ways of expediting oral mucosal wound healing.

Oral mucosal wound healing involves the following stages: the hemostasis phase, the inflammation phase, the proliferation phase, and the tissue remodeling phase [6]. The phases involve inflammation, re-epithelialization, angiogenesis, extracellular matrix deposition, and remodeling. Several methods have been used to speed up the healing of oral mucosal wounds, such as choosing monofilament synthetic suture to reduce the risk of infection [7], using low-level laser therapy to reduce inflammation and edema [8,9], and applying topical gel that contains pro-healing compounds [10,11]. It is possible to speed up the wound-healing process in several ways. Re-epithelialization is an important step and a crucial parameter of wound healing, which depends mainly on keratinocyte expansion and migration [12,13]. Therefore, the migration, proliferation, and differentiation of keratinocytes are crucial for effective re-epithelialization and restoring the barrier function of the oral mucosa. Many modulators are involved in re-epithelialization, including growth factors, cytokines, matrix metalloproteinases, and extracellular matrix components [14,15]. Studies have demonstrated that Mediator complex subunit 1(MED1) regulated the proliferation and differentiation of keratinocytes in the skin and played a key role in epidermal homeostasis [16]. Furthermore, 8-week-old keratinocyte-specific MED1-null (Med1^epi−/−^) mice showed faster skin wound healing than age-matched wild-type mice [17].

MED1 is one of the important subunits of the Mediator complex. Research studies supported the idea that MED1 played an important role in cell proliferation, differentiation, metabolism, and internal environmental stability [18,19,20,21]. MED1 is involved in regulating the expression of follistatin. Follistatin expression in the skin and liver of MED1-specific ablation mice was significantly decreased [17,22]. Follistatin is a natural antagonist of activins. Two follistatin molecules bind to one activin molecule and neutralize its ligand, resulting in antagonism of the activin receptor [23].

Activins belong to the transforming growth factor (TGF)-β superfamily. According to the different binding forms of its subunits, activins form three molecular structures, activin A, activin B, and activin AB, which have similar biological functions [24]. Activins are the main activators of activin receptors and downstream signaling cascades. Activins signal through a family of transmembrane serine/threonine kinase receptors [25]. They bind with type II Activin receptors (ActRII or ActRIIB), causing recruitment, phosphorylation, and activation of type I Activin receptors, which can activate two signaling pathways: the Smad signaling pathway and the mitogen-activated protein kinase (MAPK) signaling pathway. The latter mainly includes extracellular signal-regulated protein kinase (ERK), C-Jun N-terminal kinase (JNK)/stress-activated protein kinase (SAPK), p38 MAPK, and ERK5/BMK1 pathways (Appendix A). Each subfamily could be activated by different external factors, conduct different pathways, and mediate different biological effects. During wound healing, extracellular matrix deposition and fibrosis are primarily controlled by Smad signaling [26,27]. Zong et al. showed that ERK signaling was involved in angiogenesis and promoted skin wound healing [28]. P38 kinase cascades played an important role in wound contraction according to Hirano et al. [29]. Zhang et al. [30] found that activin B promoted the proliferation of keratinocytes and hair follicle cells at the wound area and promoted wound closure in vivo through the RhoA-Rock-JNK-cJun signaling pathway. Another study revealed that activation of JNK and ERK signaling were required for activin B induced cell migration [31]. Additionally, inhibition of JNK activity promoted differentiation of epidermal keratinocytes and enhanced wound healing according to Gazel et al. [32]. These findings show that the JNK signaling pathway is related to keratinocyte proliferation, migration, and differentiation.

According to the above information, we hypothesize that MED1 may also play a role in wound healing of the oral mucosa. As a result of MED1 ablation, follistatin expression was decreased, activins were elevated, and the JNK signaling pathway was strongly activated. Transgenic mice have become an important tool for studying gene function and modeling human diseases in vivo. Therefore, our current studies validate this hypothesis through MED1 ablation mice in vivo and human oral keratinocytes for in vitro experiments.

## 2. Results

### 2.1. Thickened Oral Mucosal Epithelial Layer in Med1^epi−/−^ Mice

The construction of Med1^epi−/−^ mice is shown in Figure 1A. MED1 *loxP/loxP* mice were intercrossed with transgenic mice expressing K14 Cre recombinase under the control of K14 promoter. Compound heterozygotes (K14 Cre +/−, MED1 +/*loxP*) were then crossed to homozygous MED1 *loxP/loxP* mice to generate MED1 KO mice (K14 Cre +/−, MED1 *loxP/loxP,* that is Med1^epi−/−^ mice) and their control littermates (K14 Cre −/−, MED1 *loxP/loxP*). The genotyping diagram of Med1^epi−/−^ mice are shown in Figure 1B. Cre negative (-) and *loxP* positive (+/+) mice were used as the control group. Cre positive (+) and *loxP* positive (+/+) mice were selected as the experimental group. Based on immunohistochemical staining, the MED1 gene was specifically ablated in the oral mucosal epithelium of Med1^epi−/−^ mice. The brownish-black nucleus indicates positive MED1 expression. As shown in Figure 1C, the brown or tan granules (indicated by black arrows) were present in the nucleus of the oral mucosal epithelial layer in the control group but not in the KO group. These results demonstrated that the MED1 protein was missing from keratinocytes of the oral mucosa in Med1^epi−/−^ mice. Moreover, we also found that the epithelial thickness of the Med1^epi−/−^ mice was thicker than that of the control mice (Figure 1D,E). This indicated that MED1 had a certain effect on keratinocyte proliferation when oral mucosa was not injured.

### 2.2. Ablation of MED1 Accelerated Oral Mucosal Wound Healing

Macroscopic evaluation on day 7 revealed that the wound of the MED1 ablation group had completely healed earlier than that of control group (Figure 2A). On days 3, 5, and 7, the open wound area in the knockout group was significantly smaller than that in the control group (Figure 2B). After the injury, the distance between the edges of the epithelium was gradually shortened, as shown by hematoxylin and eosin (H–E) staining of wound sites (Figure 2C). On day 5 and day 7, the distance between the migrating epithelium of the Med1^epi−/−^ mice was significantly shorter than the control group (Figure 2D). In the skin wound model, 8-week-old Med1^epi−/−^ mice showed an increased rate of re-epithelialization of the wound and accelerated proliferation of keratinocytes, while the expression of myofibroblasts in granulation tissue is unchanged [17]. The study showed that MED1 knockdown affected epidermal cell function but had no effect on granulation or scar formation. In the oral wound model, we also found that the healing speed of oral mucosal wound in Med1^epi−/−^ mice at 8 weeks was faster than that in the control mice, and H–E staining showed that the process of post-trauma re-epithelialization was accelerated, indicating that epithelial knockout of MED1 could also promote the healing of oral mucosal wound.

### 2.3. Ablation of MED1 Accelerated Proliferation and Migration of Keratinocytes in Mice

To determine the mechanism underlying the accelerated wound healing in Med1^epi−/−^ mice, we next compared the proliferation of keratinocytes by Ki67 immunostaining of the wound mucosa. Ki67 is a nuclear antigen associated with cell proliferation and is present throughout the cell cycle. Therefore, immunostaining with Ki67 monoclonal antibody is an effective method to evaluate cell growth fraction [33]. The results showed that the number of Ki67-positive keratinocytes in Med1^epi−/−^ mice was higher than that in the control mice on days 1, 3, 5, and 7 after wound construction (Figure 3A,B), indicating a higher proliferation rate of keratinocytes in Med1^epi−/−^ mice. Next, we performed the uPAR immunostaining of the mucosal wound to assess keratinocyte migration activity. uPAR is an important regulator of extracellular matrix proteolysis, cell-extracellular matrix interactions, and intercellular signaling [34]. uPAR can also promote cell motility, invasion, proliferation and survival. In particular, uPAR is expressed only in certain tissues, such as pregnancy tissue during embryo implantation and placental development [35], central nervous system after ischemia or trauma [36], and keratinocytes during epidermal wound healing [37]. Therefore, uPAR can generally be regarded as a marker of keratinocyte migration [38]. Our results showed that the number of uPAR-positive keratinocytes was higher on the day 1 and day 3 after trauma, and the proportion of uPAR-positive keratinocytes was higher in the MED1 ablation group as compared to the control group on days 1 and 3 (Figure 3C,D). These results indicated that keratinocyte migration was more active in the oral mucosa of mice in the early stage of wound repair, and MED1 knockdown promoted the migration of oral keratinocytes at this stage. The number of uPAR-positive keratinocytes was relatively low on day 5 and day 7 after trauma. Additionally, there was no significant difference between the MED1 ablation group and control group (Figure 3C,D). These results indicated that keratinocyte migration was relatively less in the late stage of wound repair, which was consistent with the migration pattern of keratinocyte during wound repair [39].

### 2.4. Keratinocytes with MED1 Knockdown Exhibit Enhanced Proliferation and Migration In Vitro

As further validation of in vivo results, we established the MED1-knockdown human oral keratinocyte (HOK) in vitro using lentivirus transfection technology. Three RNAi sequences targeting for MED1 were designed with the Invitrogen RNAi Designer. MED1 shRNA lentiviral vectors (Sh-MED1-1, Sh-MED1-2 and Sh-MED1-3) were created by inserting the sequence targeting MED1 into the GV493 lentiviral vector. GFP was detected by fluorescence microscopy (Figure 4A). After the knockdown, the level of MED1 gene expression was decreased significantly as indicated by qRT-PCR (Figure 4B). The Sh-MED1-2 group showed the highest transfection efficiency and was therefore selected for further experiments. Western blot demonstrated that MED1 protein expression was also significantly downregulated in this group (Figure 4C). To evaluate the effect of MED1 knockdown on cell proliferation activity, a CCK-8 assay was performed. The results revealed that MED1 knockdown markedly promoted HOK proliferation (Figure 4D). Next, a scratch wound assay demonstrated that MED1-knockdown HOK exhibited higher migration ability compared with the control group (Figure 4E,F).

### 2.5. Follistatin Expression Is Decreased and the JNK/c-Jun Pathway Is Activated in MED1 Knockdown Keratinocytes In Vitro

To define the specific role that MED1 plays in wound healing, we examined the expression levels of follistatin and activin A. qRT-PCR and Western blot confirmed that follistatin expression was significantly decreased and activin A expression was increased in MED1 knockdown keratinocytes compared with the control (Figure 5A,B). There was no significant difference between the control and knockdown groups in total JNK, ERK and p38 mRNA expression. However, MED1 knockdown keratinocytes expressed higher levels of total c-Jun (a representative target of JNK substrates) as identified through mRNA expression (Figure 5C). Meanwhile, the phosphorylation of JNK as well as c-Jun was enhanced in MED1 knockdown keratinocytes compared with the control group while the phosphorylation of ERK and p38 was not enhanced (Figure 5D). To further validate the molecular mechanism, sh-MED1 cells were treated with JNK inhibitor SP600125. CCK-8 assay showed proliferation ability was significantly lower in the Sh-MED1 + SP600125 group (Figure 5E). Scratch wound assay showed that the addition of SP600125 inhibited cell migration (Figure 5F,G). The results demonstrated that the JNK/c-Jun signaling pathway was involved in the proliferation and migration of MED1 knockdown keratinocytes.

## 3. Discussion

This study revealed that 8-week-old mice carrying the Med1^epi−/−^ mutation exhibited accelerated wound healing compared to age-matched control mice. H–E staining results indicated that re-epithelialization was faster in Med1^epi−/−^ mice. Based on immunostainings of the traumatized epithelium, Med1^epi−/−^ mice showed enhanced proliferation and migration of keratinocytes after the wound. During the healing process, the keratinocytes switch to a migratory mode and migrate into the damaged epithelium to fill it in, then new keratinocytes proliferate to replace the migrating cells. The migration and proliferation of keratinocytes are critical for effective re-epithelialization and wound healing [39,40]. We found that MED1 ablation promotes wound healing by increasing keratinocyte proliferation and mobility in the oral wound-healing process.

MED1 is a large multifunctional protein that resulted in embryonic lethality at around day 11 when whole-body knockouts of the MED1 gene are performed in mice. To study MED1’s biological role in specific tissues, tissue-specific conditional knockout mice have been generated. Epithelial-specific MED1 null mice exhibited abnormal hair differentiation and cycling [16]. The Med1^epi−/−^ mice showed ectopic hair on the labial of the incisor and enamel hypoplasia (insufficient enamel mineralization) in the incisor [41], indicating that ablation of MED1 switches the cell fate of dental epithelia to that generating hair. We have found that MED1 participates in the re-epithelialization of the oral epithelium. These results suggest that MED1 plays an equally important role in the function of epidermal/epithelial cells. As a result, the underlying mechanism should be investigated.

It was found that the knockdown of MED1 significantly reduced the expression of follistatin in human oral keratinocytes. This indicates that ablation of MED1 suppresses the expression of follistatin, similar to what has been previously reported [22]. As a multifunctional regulatory protein, follistatin is expressed to varying degrees in multiple tissues, and its expression level is regulated by a variety of factors, such as transcription factors, oxidative stress level, and mechanical stress [42]. As a natural antagonist of activins, follistatin can bind at a site of activins with high affinity and block their receptor activation and signal transduction.

Studies have shown that activins play a crucial role in wound healing and they are involved in epidermal re-epithelialization, keratinocyte proliferation, granulation tissue formation, and collagen synthesis by fibroblasts [25]. The activin-overexpressing transgenic mice showed accelerated skin wound healing with enhanced re-epithelialization [43]. Interestingly, the mice overexpressed with follistatin in keratinocytes displayed a contrasting phenotype. These follistatin-overexpressing mice were characterized by slowed re-epithelialization and a severe delay in wound repair [44]. This suggests that the abnormalities observed in mice overexpressing follistatin are indeed caused by the inhibition of endogenous activins. The balance of the activin/follistatin system is crucial for maintaining homeostasis of the internal environment, and once the balance is broken, the related signal transduction would change, which resulted in different biological effects. The expression level of activins is generally low in normal adult tissues. Activin A gene expression is elevated after the injury and localized in keratinocytes, while activin B gene expression is lower and localized in the hyperproliferative epithelium at the wound margin [45]. Therefore, our experiment focused on the effect of MED1 knockdown on the expression of activin A.

Activin A was found to be overexpressed in MED1 knockdown oral keratinocytes in the current study. Additionally, the JNK/c-Jun pathway was activated in MED1 knockdown keratinocytes, while ERK and the p38 pathway were not activated. A JNK inhibitor significantly inhibited MED1 knockdown-induced proliferation and migration of oral keratinocytes, suggesting that the JNK/c-Jun signaling pathway is involved in this process.

As a result, we propose that the MED1 ablation actions are similar to relieving the “molecular brake” (just as when a foot is off the brake pedal) on keratinocytes and change the activin A/follistatin system to activate the JNK/c-Jun signaling pathway in keratinocytes more effectively. The proliferation and migration of keratinocytes are enhanced after MED1 knockdown, thereby promoting re-epithelialization, which helps to accelerate the healing process of oral mucosal wounds (Figure 5H).

Rapid wound healing could reduce the risk of infection and prevent oral functional impairment such as dysphagia, dysphonia, and masticatory difficulty. Current clinical applications for oral wound healing are relatively focused on the use of appropriate surgical sutures and laser therapy to promote the healing of oral mucosal wounds. Although these methods have been widely applied, there are still some limitations. For example, bacteria can adhere to the surface of suture materials and sutures are also a major source of stress for patients [46]. Laser therapy has strict indications and should not be used in pregnant women and patients with photosensitive epilepsy [9]. Meanwhile, different from the methods above, intervention with MED1 is rooted in the process of mucosa formation itself. It showed a wider range of applicability in wound healing.

Interestingly, MED1 deficiency accelerated skin wound healing in 8-week-old mice but delayed it in 6-month-old mice. It may due to the depletion of hair follicle stem cells in old mice, the positive effect of follistatin down-regulation on epidermal regeneration was overcome and eventually resulted in impaired cutaneous wound healing [17]. Although we did not investigate the effect of aging on oral mucosal wound healing in Med1^epi−/−^ mice, we believe this effect is limited. This is largely due to the distinct differences in the genomics and dynamics of the healing process in the skin and oral mucosa [47,48]. The hair follicle is a cutaneous appendage that houses multiple epithelial stem cells [49], while the oral mucosa has no hair follicle structure, and its epithelial stem cells are mostly located in the basal layer [50]. In addition, saliva contains numerous and diverse cytokines, growth factors, histones, antimicrobial peptides and many more constituents, which can promote the proliferation and migration of keratinocytes or fibroblasts [51,52,53]. This may mitigate, to some extent, the effects of aging on oral mucosal wound healing in Med1^epi−/−^ mice. Certainly, further experiments are necessary to confirm this. In addition, the mechanisms which contributed to follistatin downregulation in MED1 knockdown keratinocytes also require further investigation. Future studies will focus on the relationship between MED1 and follistatin. Considering that MED1 acts as a transcriptional co-activator, we propose that MED1 may activate the follistatin promoter and stimulate follistatin transcription. That is, MED1 could regulate the expression of follistatin at the transcriptional level and thus result in cascade reactions after MED1 knockdown in keratinocytes, including the activated of activin A and JNK/c-Jun signaling. Future studies will be required to experimentally test this hypothesis.

In addition, the relationship between MED1 and immune function should not be ignored. Yue et al. have demonstrated that T-cell–specific ablation of MED1 specifically impairs the generation of invariant natural killer T-cell (iNKT) [54]. iNKT cells are a unique lineage of T lymphocytes that regulate both innate and adaptive immunity. Undoubtedly, innate and adaptive immunity play essential roles in wound healing [55,56]. Bai et al. have generated macrophage-specific MED1/apolipoprotein E (ApoE) double-deficient (MED1^ΔMac^/ApoE^−/−^) mice and found that MED1 ablation decreased the binding of peroxisome proliferator-activated receptor γ (PPARγ), which promoted the polarization of macrophages from M2 to M1 and enhanced innate immune stimulation, thereby increasing atherosclerosis [57]. Therefore, MED1 could regulate the M1/M2 phenotype switch of macrophage and exerted the anti-inflammatory effect on macrophages. Given that MED1-regulated gene expression is tissue-dependent and cell-dependent, our future research will focus on whether MED1 would also regulate the expression of immune cells in the oral mucosa. To summarize, the wound healing and re-epithelialization is a complex and precisely regulated process. The possibility that other regulatory factors contribute to the healing process of the wound in MED1 ablation oral mucosa cannot be ruled out. These possibilities all deserve further investigation.

Although the mice models do not replicate human disease in its entirety, they have contributed to our understanding of mechanisms underlying human oral mucosal wound healing. Several current studies and modalities have been developed for promoting oral wound healing, including polymeric stents, biological matrices, and gel-like ointments. It is also possible to combine drugs, proteins, cells, tissue, or growth factors with the delivery vehicles to enhance therapeutic efficacy. It is promising that MED1 inhibitors can be loaded into mucosal delivery vehicles by physical loading or chemical conjugation, thereby providing it with therapeutic potential.

## 4. Materials and Methods

### 4.1. Laboratory Animals

All animal studies were approved by the Animal Care and Use Committee for Tianjin Medical University, in compliance with the Guide for the U.S. Public Health Service’s policy on humane care and use of laboratory animals. Animals were housed with 12 h light/dark cycles and received food, standard rodent chow, and water ad libitum in compliance with the Association for Assessment and Accreditation of Laboratory Animal Care International guidelines.

### 4.2. Generation of Keratinocyte-Specific Med1^epi−/−^ Mice

We have generated conditional Med1^epi−/−^ mice as described previously [16]. For this, conditional MED1 ablation mice were disrupted under control of the keratin 14 (Krt14) promoter. Floxed MED1 mice (C57/BL6 background) were mated with transgenic mice expressing Cre recombinase under the control of the Krt14 promoter. A total of 72 female mice (avoiding the effects of sex hormones) that weight about 18 g to 25 g were used for the experiments. We have selected 8-week-old homozygous floxed mice with the *Cre* transgene as the experimental group (KO) whereas littermates with floxed MED1 alleles but no *Cre* were chosen as the control group (CON). Each of these groups contained 36 mice and each group was divided into four within itself (9 per group). Randomization sequences were generated in advance using a computer-generated randomization program. The mice were randomly assigned to groups of predetermined sample size. Mice with different genotypes were housed in the different cages in one room. Mice were observed on daily basis and treated appropriately by institutional veterinarians whenever they show any signs of discomfort or disease.

### 4.3. Oral Mucosal Injury Mice Model Construction

The mice were anesthetized with freshly prepared sodium pentobarbital (0.6%), injected intraperitoneally as well as the oral cavity was sterilized using 0.12% chlorhexidine with individual and disposable cotton. A wound in the cheek mucosa was made using a circular biopsy punch of 3 mm diameter. The operation was carried out by the same person. We placed the animals in a thermic bag to avoid hypothermia and observed them during the return of the anesthetic plane. The mice were fed a soft diet following model establishment. The wound healing was macroscopically monitored by digital photography at day 1, 3, 5, and 7 after the wound. On each wound margin, transparent film was used to depict the wound margin, and the long diameter and short diameter of the wound were measured by a vernier caliper. The open wound area is calculated with the formula: long diameter × short diameter × π/4. The data analysis was performed by the second author. The animals were euthanized at the specified time point. We collected wound-containing tissues, fixed them with 4% paraformaldehyde, embedded them in paraffin, and then sectioned them.

### 4.4. Analysis of Cell Migration in Wound Tissue

The migration capacity of keratinocytes is determined using urokinase plasminogen activator receptor (uPAR, a marker of keratinocytes migration [39]) through immunofluorescence staining technique. Anti-uPAR mouse antibodies (sc-13522, 1:150, Santa Cruz Biotechnology, Santa Cruz, CA, USA) were incubated with oral mucosa sections, followed by detection using Alexa 568 anti-mouse secondary antibody. DAPI (4,6-diamidino-2-phenylindole) was used to label nuclei. Images were obtained with a fluorescence microscope.

### 4.5. Analysis of Cell Proliferation in Wound Tissue

The proliferative capacity of keratinocytes was determined by Ki67 (a hallmark of keratinocyte proliferation) immunohistochemical staining. Then, 5 mm thick paraffin sections contained traumatized tissue were deparaffinized and autoclaved in sodium citrate buffer (10 mmol/L sodium citrate, pH 6.0) for 15 min, then microwaved to repair the antigens. After washing in PBS-T (Phosphate Buffered Saline with 1% Tween 80), the sections were treated with H_2_O_2_ to block the endogenous peroxidase activity. This was followed by blocking with 5% goat serum (Thermo Fisher Scientific, Waltham, MA, USA) at room temperature for 60 min. The primary antibodies (rabbit anti-Ki67, ab197234, 1:100, Abcam, Cambridge, UK) were incubated at 4 °C overnight. Goat anti-rabbit horseradish peroxidase (HRP)-conjugated antibody (RS001,1:400, Immunoway, Plano, TX, USA) was used as the secondary antibody in the study. The staining was revealed with diaminobenzidine (DAB) for 3–5 min. Hematoxylin staining was performed at room temperature for 10 s. The sections were then washed in PBS-T for 5 min to return them to blue, followed by dehydration and neutral resin packing. A vertical microscope was used to observe positive images.

### 4.6. Production and Culture of MED1 Knockdown Keratinocytes

We chose human oral keratinocytes (HOK, purchased from Tongpai Biotechnology Co., Ltd, Shanghai, China) for the in vitro study. The cells were maintained in cultured in defined keratinocyte serum-free medium (Defined keratinocyte-SFM, Gibco, Billings, MT, USA). The RNAi target sequences are as follows: MED1 RNAi 1, gcCGAGTTCCTCTTATCCTAA; MED1 RNAi 2, ccTTTATGGAAGCAGCCCTTT; RNAi 3, gcTCTCAAAGTAACATCTTTA; and negative control RNAi, TTCTCCGAACGTGTCACGT. The lentivirus vectors constructed according to the above target sequences were provided by Genechem Chemical Technology Co., Ltd (Genechem, Shanghai, China) Cells transduced with the empty vector acted as the negative control. Green fluorescent protein (GFP) was labeled on all lentiviral vectors. The transfection was performed following the manufacturer’s instructions. Cells were harvested 48–72 h post-transfection. Transfection efficiency was determined by quantitative real-time PCR (qRT-PCR) and Western blot analysis.

### 4.7. Total RNA Extract and qRT-PCR

Total RNA was isolated from cells using the Trizol reagent (Invitrogen, Carlsbad, CA USA). cDNA was obtained by reverse transcription reactions using the reverse transcription kit (Applied Biosystems, Foster, CA, USA). To perform real-time quantitative PCR (qRT-PCR), we use iTaq Universal SYBR Green Supermix (Bio-Rad, Hercules, CA, USA) and the 7500HT Fast Real-Time PCR System (Applied Biosystems, Foster, CA, USA). All primers used in the study are listed in Appendix A. The relative abundance of mRNA was normalized to GAPDH.

### 4.8. Immunoblotting

Cells were seeded in 6-well plates. Cell lysates used for immunoblotting were prepared in RIPA buffer (Abcam, Cambridge, UK). Protein concentrations were analyzed using the BCA Protein Assay Kit (Thermos Fisher Scientific, Waltham, MA, USA). Then, 20~25 μg of different proteins were loaded on lanes and separated by SDS-PAGE and transferred to nitrocellulose membrane (Bio-Rad, Hercules, CA, USA). After blocking with 5% defatted milk (EpiZyme Biotechnology, Shanghai, China), membranes were incubated with a primary monoclonal antibody against MED1, Follistatin, activin-A, JNK, phospho-JNK, c-Jun, phospho-c-Jun, and β-actin at 4 °C overnight. The membranes were then washed and incubated with HRP-conjugated secondary antibody at RT for 1 h. Protein bands were detected using a chemiluminescence system (ECL Kit) and quantified in ChemiDoc XRS Imaging System. The antibody details are shown in Appendix A.

### 4.9. Keratinocyte Proliferation Assay

Cell proliferation was detected using the *Cell Counting Kit*-*8* (CCK-8) assay. In short, a total of 2000 cells per well were inoculated into 96-well plates and cultured for 24 h. Next, 10 µL CCK-8 solution was added to each well and the cells were co-incubated at 37 °C for 2 h daily for 5 days. The OD value was measured by an enzyme labeling instrument at 450 nm wavelength.

### 4.10. Keratinocyte Migration Assay

A total of 5 × 10^5^ cells were seeded in 6-well plates and cultured until 100% confluent. The monolayer cells were scratched using a 200 μL pipette tip and the plate was washed 3 times with PBS. The scratch wounds were marked with a labelling pen and photographed. After that, DMEM supplemented with 2% FBS was added and the cells were incubated with 5% CO_2_ at 37 °C. Migration was monitored and photographed after 6 h, 12 h, and 24 h, respectively. Wound closure analysis was performed using Image J software. Wound-healing percentage was calculated as the following formula: (original scratch area—the point scratch area)/original scratch area × 100%.

### 4.11. Analysis of Cell Proliferation in Wound Tissue

The proliferative capacity of keratinocytes was determined by Ki67 (a hallmark of keratinocyte proliferation) immunohistochemical staining. Then, 5 mm thick paraffin sections contained traumatized tissue were deparaffinized and autoclaved in sodium citrate buffer (10 mmol/L sodium citrate, pH 6.0) for 15 min, then microwaved to repair the antigens. After washing in PBS-T (Phosphate Buffered Saline with 1% Tween 80), the sections were treated with H_2_O_2_ to block the endogenous peroxidase activity. This is followed by, blocking with 5% goat serum (Thermos Fisher Scientific, Waltham, MA, USA) at room temperature for 60 min. The primary antibodies (rabbit anti-Ki67, ab197234, 1:100, Abcam, Cambridge, UK) were incubated at 4 °C overnight. Goat anti-rabbit horseradish peroxidase (HRP)-conjugated antibody (RS001, 1:400, Immunoway, Plano, TX, USA) was used as the secondary antibody in the study. The staining was revealed with diaminobenzidine (DAB) for 3–5 min. Hematoxylin staining was performed at room temperature for 10 s. The sections were then washed in PBS-T for 5 min to return them to blue, followed by dehydration and neutral resin packing. A vertical microscope was used to observe positive images.

### 4.12. Inhibition Assay

20 μM JNK inhibitors SP600125 (dissolved in DMSO, MCE, Belleville, NJ, USA) were added into the media 2 h before any additional treatment. Equal amounts of DMSO (Solarbio, Beijing, China) were added in the control group.

### 4.13. Statistical Analyses

Statistical analyses were performed using Graphpad Prism 8 statistical software (GraphPad software, La Jolla, CA, USA). Group results are expressed as mean values ± SD. Data between two groups were compared using unpaired two-tailed Student’s *t-tests*, with a significance level set at *p* < 0.05. Three independent experiments were conducted.

## 5. Conclusions

In summary, our study revealed the role of the MED1 in oral mucosal wound healing. We propose that the MED1 ablation change the activin A/follistatin system and activate the JNK/c-Jun signaling pathway in keratinocytes more effectively. The activation of JNK cascade signaling pathway can enhance the proliferation and migration of keratinocytes, thereby promoting re-epithelialization, and thus helping to accelerate the healing process of oral mucosal wound.

## Figures and Tables

**Figure 1 ijms-23-13414-f001:**
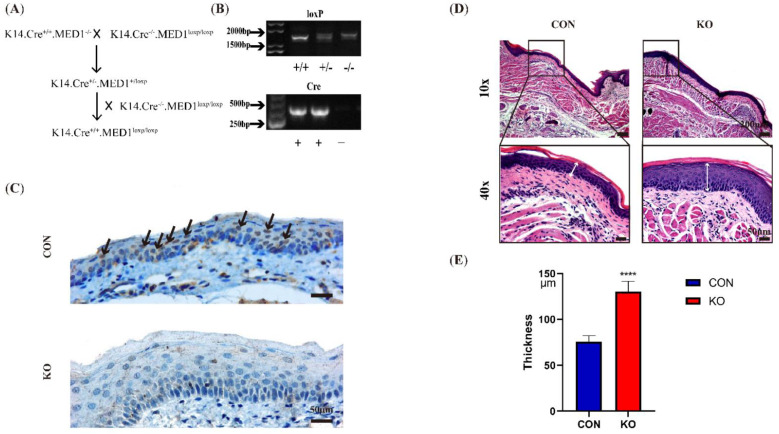
The construction of Med1^epi−/−^ mice and the thickness of oral mucosa epithelium. (**A**) The construction diagram of Med1^epi−/−^ mice, (**B**) Electrophoresis results of loxP primers and Cre primers. The Genotypes were determined by PCR using genomic DNA derived from MED1 heterozygous mutant, MED1 homozygous mutant, and control mice. (+/+) represented homozygous positive type, (+/−) heterozygous type and (−/−) homozygous negative type in loxP electrophoretic diagram, (+) represented positive type and (−) represented negative type in Cre electrophoretic diagram, (**C**) Immunohistochemical staining of MED1 in oral mucosa epithelium of control mice (CON) and Med1^epi−/−^ mice (KO) before modeling. Black arrowheads indicate MED1-positive cells, (**D**,**E**) H–E staining images of the oral mucosa and analysis of oral mucosal epithelial thickness. White arrowheads represent the epithelial thickness. Scale bar = 200 μm (10 × and 50 μm (40 ×). **** *p* < 0.0001. All data were presented as means ± SD.

**Figure 2 ijms-23-13414-f002:**
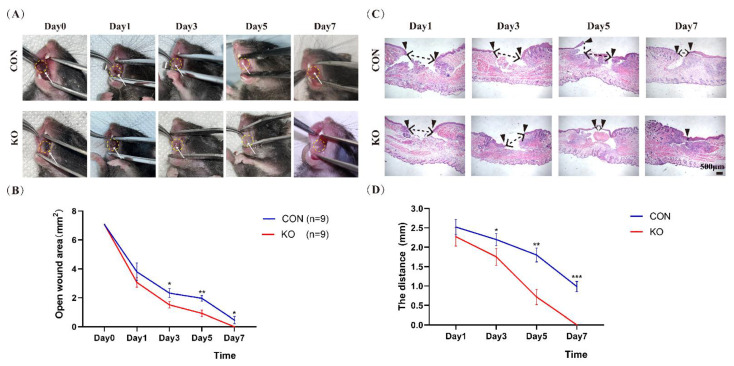
Oral wound healing is accelerated in Med1^epi−/−^ mice. (**A**) Representative macroscopic views of oral wounds on days 1, 3, 5, and 7 after injury in control and MED1 null mice. The yellow dotted circles indicate the initial position of the wound. Trauma locations are indicated with white arrows, (**B**) Evaluation of open wound area by morphometrical analysis of the wound areas. n = number of measurements, (**C**) Representative histological view of oral mucosal re-epithelialization. Black dotted lines indicate the distance between the edges of the epithelium, (**D**) The distance between wound margins. Scale bar = 500 μm. Bars = means ± SD. * *p* < 0.05, ** *p* < 0.01.*** *p* < 0.001.

**Figure 3 ijms-23-13414-f003:**
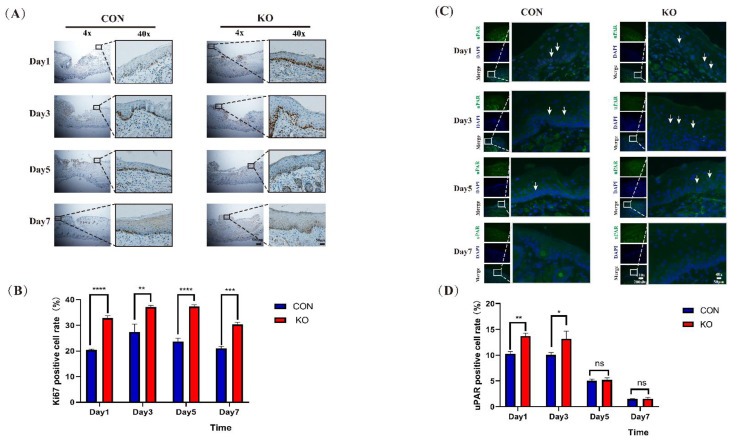
The proliferation and migration of keratinocytes were enhanced in Med1^epi−/−^ mice. (**A**) Representative mucosal sections showing immunohistochemical staining for Ki67, (**B**) Quantification of the number of Ki67-positive cells, (**C**) Immunofluorescence staining of uPAR for the detection of keratinocyte migration, (**D**) Quantification of the number of uPAR-positive cells. Scale bar = 500 μm (4 ×), 200 μm (10 ×), and 50 μm (40 ×). Bars = means ± SD. ns means nonsignificant, * *p* < 0.05, ** *p* < 0.01, *** *p* < 0.001, **** *p* < 0.0001.

**Figure 4 ijms-23-13414-f004:**
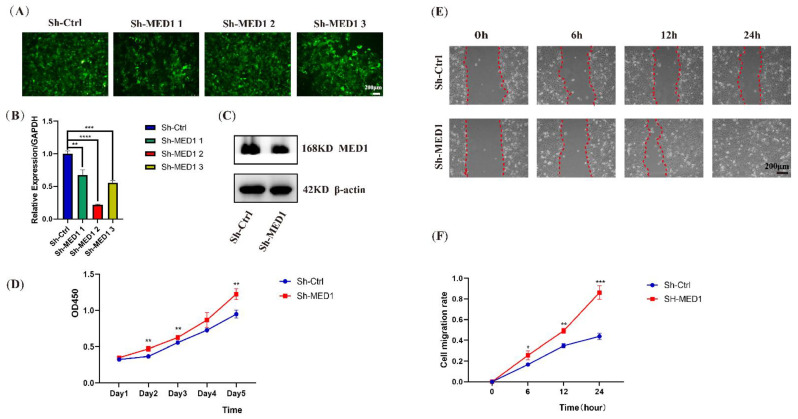
MED1 knockdown promotes human oral keratinocytes proliferation and migration in vitro. (**A**) Human oral keratinocytes (HOK) expressing green fluorescent protein (GFP) could be observed in MED1 lentivirus transfected group and the negative control group under fluorescence microscopy, (**B**) MED1 mRNA expression levels in keratinocytes after 72 h of lentivirus transfection, (**C**) Expression of MED1 protein after MED1 knockdown in keratinocytes, (**D**) Cell proliferation of transfected HOK using CCK-8 assay, (**E**) The typical pictures of scratch wound assay for transfected HOK, (**F**) Quantification relative migration rate after initial scratch time for transfected HOK. Scale bar = 200 μm. Bars = means ± SD. * *p* < 0.05, ** *p* < 0.01, *** *p* < 0.001, **** *p* < 0.0001.

**Figure 5 ijms-23-13414-f005:**
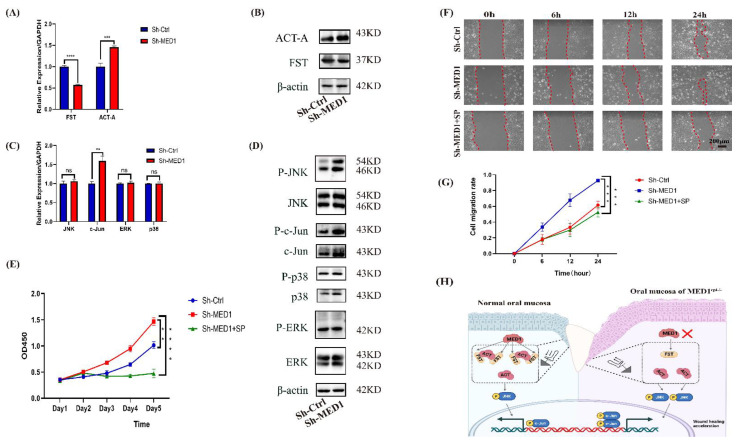
Activin-A/follistatin system is changed and the JNK/c-Jun pathway is activated in MED1 knockdown keratinocytes. (**A**) Relative expression of mRNA encoding for FST and ACT-A following transfection in HOK, (**B**) Relative expression of protein for FST and ACT-A following transfection in HOK, (**C**) Relative mRNA expression of JNK, c-Jun, ERK and p38, (**D**) Western blot assays for c-Jun, P-c-Jun, JNK, P-JNK, ERK, P-ERK, p38 and P-p38 in HOK after transfection, (**E**) The results of CCK-8 assay for sh-MED1 cells following the addition of SP600125, (**F**) The results of scratch wound assays for sh-MED1 cells following the addition of SP600125, and (**G**) Quantitative analysis of migrated cells, (**H**) Proposed model of accelerated oral mucosal wound healing in Med1^epi−/−^ mice. Scale bar = 200 μm. Bars = means ± SD. ns means nonsignificant, * *p* < 0.05, ** *p* < 0.01, *** *p* < 0.001, **** *p* < 0.0001.

## Data Availability

Not applicable.

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
