# Peer review of "MED1 Ablation Promotes Oral Mucosal Wound Healing via JNK Signaling Pathway"

_ijms, 2022, doi:10.3390/ijms232113414_

Round 1

Reviewer 1 Report

In the manuscript by Meng et al., the authors describe how disrupting expression of MED1 accelerates mucosal wound healing. This is a straightforward study of fairly limited scope that presents what appears to be an unexpected set of results that suggest genetic deletion or shRNA mediated silencing of MED1 hastens mucosal wound healing. Strengths of the study include a use of in vitro and animal studies, as well as complementary approaches in genetic deletion and shRNA in murine and human systems, respectively. A number of concerns require attention before the manuscript is fit for publication, and these are listed below.

1.     The manuscript is in a preliminary state of preparation. In particular, extensive English language editing is needed to correct the many grammar issues that detract from the overall work and, more importantly, hinder clarity at points. These include prevalent missing words, incorrect tenses, redundant/awkward phrasing, and an overly colloquial tone in places.

2.     Vague and non-specific statements like that made on lines 74-75 (“JNK signaling pathway is related to keratinocyte proliferation…”) should be revisited to provide more informative and scholarly content

3.     Some results sections are lacking in the rationale for the experiments performed as well as the implications of the findings (particularly true for sections 2.1-2.3). Descriptions of presented figures are overly brief and missing in some cases (e.g., Figure 2D).

4.     Figure 2B: It is indicated that the control and knockout groups are significantly different  on the day 7 time point. This seems highly unlikely given the near overlapping positions of the plotted means. Differences in wound area shown in this figure appear barely different. It is also unclear what the capital and lowercase “n” values in the figure key represent.

Author Response

Reviewer 1

In the manuscript by Meng et al., the authors describe how disrupting expression of MED1 accelerates mucosal wound healing. This is a straightforward study of fairly limited scope that presents what appears to be an unexpected set of results that suggest genetic deletion or shRNA mediated silencing of MED1 hastens mucosal wound healing. Strengths of the study include a use of in vitro and animal studies, as well as complementary approaches in genetic deletion and shRNA in murine and human systems, respectively. A number of concerns require attention before the manuscript is fit for publication, and these are listed below.

  1. The manuscript is in a preliminary state of preparation. In particular, extensive English language editing is needed to correct the many grammar issues that detract from the overall work and, more importantly, hinder clarity at points. These include prevalent missing words, incorrect tenses, redundant/awkward phrasing, and an overly colloquial tone in places.

Response: We would like to express our heartfelt thanks to reviewer 1 for his thorough reading of our manuscript and for his very useful suggestions. We have thoroughly revised our manuscript. During the revision process, we have also done a very careful language polishing guided by a native speaker.

  1. Vague and non-specific statements like that made on lines 74-75 (“JNK signaling pathway is related to keratinocyte proliferation…”) should be revisited to provide more informative and scholarly content.

Response: Thanks for your detailed review. We have included a more detailed description of the function of JNK signaling pathway in the wound healing progress. The revision is on lines 76-81 and has been marked in Yellow.

  1. Some results sections are lacking in the rationale for the experiments performed as well as the implications of the findings (particularly true for sections 2.1-2.3). Descriptions of presented figures are overly brief and missing in some cases (e.g., Figure 2D).

Response: Thanks for your review. We have included relevant experimental principles and the significance of our results in revised manuscript. Additionally, we also described the figures in more detail as requested. The revisions are on sections 2.1-2.3 and have been marked in Yellow.

  1. Figure 2B: It is indicated that the control and knockout groups are significantly different on the day 7 time point. This seems highly unlikely given the near overlapping positions of the plotted means. Differences in wound area shown in this figure appear barely different. It is also unclear what the capital and lowercase “n” values in the figure key represent.

Response: We thank the reviewer for pointing out this problem. We corrected the data of Figure 2B by increasing numbers of mice. Correspondingly, we have replaced the images in Fig. 2A with more representative pictures. Besides, we added the immediate post-trauma pictures of two groups and indicated the initial position of the wound with yellow dotted circles to make the contrast clearer. The capital and lowercase “n” values in the figure has been unified as lowercase “n” to represent the number of mice.  

Reviewer 2 Report

In this work, the role of Mediator complex subunit 1 (MED1) in promoting oral mucosal wound healing was investigated. It has been found that its regulating mechanisms are related to the C-Jun N-terminal kinase (JNK) signaling pathway, which induces keratinocytes behaviors toward re-epithelialization. Overall, the manuscript is scientifically well-sounded together with representative results. However, some issues are raised by the reviewer to improve its quality before publication. Here are the comments:

1.        Macroscopic images in Figure 2A are somewhat unclear to represent oral wounds in a time-dependent manner. Please consider improving the image quality.

2.        In Figure 3D, please also indicate the non-significant results with “n.s.” symbol and reflect this comment to other statistical data.

3.        What are the differences between Sh-MED1-1, Sh-MED1-2, and Sh-MED1-3 that have been used for transfecting human oral keratinocytes (HOK)? Please briefly explain it in section 2.4.

4.        Figure 5 legend is missing the statistical symbol explanation. Please revise it accordingly.

5.        The authors discussed the necessity of revealing the exact mechanisms contributing to follistatin downregulation in MED1 knockdown keratinocytes. What research idea can be suggested for further investigations? It would be an excellent future perspective of this work.

6.        Superscript and subscript errors were found in section 4.10. The cell number should be 5×105 cells, while ‘CO2’ should be ‘CO2’. Please fix it.

7.        The conclusion part should be improved as the content is too short. Perhaps the authors can explain more about Figure 5H in interesting ways, or emphasize further investigations that can address unelucidated points in this current work (i.e., mechanism of MED1 knockdown regulation in oral mucosal wound healing).

8.         To the best of knowledge, this work is submitted to section molecular immunology. However, it seems that there is no clear discussion relating to immune repair or something along the immunology. Please kindly consider about this concern.

Author Response

Reviewer 2

In this work, the role of Mediator complex subunit 1 (MED1) in promoting oral mucosal wound healing was investigated. It has been found that its regulating mechanisms are related to the C-Jun N-terminal kinase (JNK) signaling pathway, which induces keratinocytes behaviors toward re-epithelialization. Overall, the manuscript is scientifically well-sounded together with representative results. However, some issues are raised by the reviewer to improve its quality before publication. Here are the comments:

  1. Macroscopic images in Figure 2A are somewhat unclear to represent oral wounds in a time-dependent manner. Please consider improving the image quality.

Response: Thank you for your valuable suggestions. We have replaced the images in Fig.2A with more representative pictures.

  1. In Figure 3D, please also indicate the non-significant results with “n.s.” symbol and reflect this comment to other statistical data.

Response: Thanks for your kind reminder. We are sorry for our negligence and “n.s” has been indicated in Figure 3D.

  1. What are the differences between Sh-MED1-1, Sh-MED1-2, and Sh-MED1-3 that have been used for transfecting human oral keratinocytes (HOK)? Please briefly explain it in section 2.4.

Response: Thanks for your kind reminder. We apologize for not explaining the differences between Sh-MED1-1, Sh-MED1-2, and Sh-MED1-3. We designed three RNAi target sequences for MED1(#1, #2, and #3) with the Invitrogen RNAi De-signer. And the three MED1 target sequences were inserted into GV493 lentiviral vectors to generate MED1 shRNA lentiviral vectors, we named them Sh-MED1-1, Sh-MED1-2, and Sh-MED1-3. The additional information has been added to the manuscript (lines 182-185 and lines 409-412) and marked in Yellow.

  1. Figure 5 legend is missing the statistical symbol explanation. Please revise it accordingly.

Response: Thanks for your kind reminder. We are sorry for our negligence and have revised it as requested.

  1. The authors discussed the necessity of revealing the exact mechanisms contributing to follistatin downregulation in MED1 knockdown keratinocytes. What research idea can be suggested for further investigations? It would be an excellent future perspective of this work.

Response: Thank you for your constructive comments. It’s our pleasure to receive your guidance. Considering that MED1 acts as a transcriptional co-activator, we propose that MED1 may activate the follistatin promoter and stimulate follistatin transcription. That is, we tentatively suggest that MED1 could regulate the expression of follistatin at the transcriptional level. Future experiments will be aimed at testing this assumption. Related discussion has also been added in the manuscript.

  1. Superscript and subscript errors were found in section 4.10. The cell number should be 5×105 cells, while ‘CO2’ should be ‘CO2’. Please fix it.

  Response: Thanks for your warm reminder. We are sorry for our negligence and have corrected it in revised manuscript.

  1. The conclusion part should be improved as the content is too short. Perhaps the authors can explain more about Figure 5H in interesting ways, or emphasize further investigations that can address unelucidated points in this current work (i.e., mechanism of MED1 knockdown regulation in oral mucosal wound healing).

 Response: Thanks for your kind reminder. We agree that the explanation of the conclusion was not thorough enough. We have now included a more detailed description of the conclusion. Revised conclusion section is marked Yellow in manuscript. We also have described and explained Figure 5H in more detail on lines 285-290. Indeed, the exact mechanism of MED1 knockdown regulation in oral mucosal wound healing is an exciting future area of investigation for our lab. As we mentioned in Question 5, we speculate that MED1 could regulate the expression of follistatin at the transcriptional level. The biological activities of activins are tightly regulated by follistatin as the binding of activin to follistatin is almost irreversible. Elevated activin-A activated the JNK/c-Jun signaling pathway and enhanced the proliferation and migration of keratinocytes after MED1 knockdown. Next work will focus on verifying whether MED1 could regulate follistatin at the transcriptional level. Related discussion has been added in the manuscript on lines 315-321.

  1. To the best of knowledge, this work is submitted to section molecular immunology. However, it seems that there is no clear discussion relating to immune repair or something along the immunology. Please kindly consider about this concern.

Response: Thank you for the question. MED1 played essential roles on immunomodulatory function(1, 2). Wound healing progress was closely related with immunomodulatory(3, 4). Therefore, whether MED1 regulated oral mucosa wound healing by immunomodulatory function is noteworthy. The detailed discussion and explanation are listed on lines 322-335. Again, we sincerely appreciate the reviewers’ valuable comments to point out another direction for our future study.

  1. Yue X, Izcue A, Borggrefe T. Essential role of Mediator subunit Med1 in invariant natural killer T-cell development. Proceedings of the National Academy of Sciences of the United States of America. 2011;108(41):17105-10.
  2. Bai L, Li Z, Li Q, Guan H, Zhao S, Liu R, et al. Mediator 1 Is Atherosclerosis Protective by Regulating Macrophage Polarization. Arterioscler Thromb Vasc Biol. 2017;37(8):1470-81.
  3. Graves DT, Milovanova TN. Mucosal Immunity and the FOXO1 Transcription Factors. Front Immunol. 2019;10:2530.
  4. Yu R, Ding Y, Zhu L, Qu Y, Zhang C, Liu L, et al. IL-22 mediates the oral mucosal wound healing via STAT3 in keratinocytes. Arch Oral Biol. 2016;72:14-20.

Round 2

Reviewer 1 Report

Thank you for making the effort to address my concerns. The manuscript reads much better now, and the passages added to the results will help the reader follow your reasoning and experimental approaches. 

-The abstract still suffers from multiple missing word errors and requires editing similar to what was applied to the rest of the work already.

-One minor tense issue was found on line 92: "The construction of Med1epi-/- mice was shown in Figure 1A." - should be changes to The construction of Med1epi-/- mice is shown in Figure 1A

After these are addressed, I feel the manuscript is ready to proceed to the publication phase.

Author Response

Thank you for making the effort to address my concerns. The manuscript reads much better now, and the passages added to the results will help the reader follow your reasoning and experimental approaches.

Author’s response: We thanks the reviewer for your efforts in reviewing our manuscript, overall positive assessment and providing constructive suggestions. All the changes were marked in GREEN in the manuscript.

-The abstract still suffers from multiple missing word errors and requires editing similar to what was applied to the rest of the work already.

Author’s response: Thanks for your review. We have checked this section thoroughly and corrected the errors accordingly. Please see line 12-25 marked in GREEN.

-One minor tense issue was found on line 92: "The construction of Med1epi-/- mice was shown in Figure 1A." - should be changes to The construction of Med1epi-/- mice is shown in Figure 1A.

Author’s response: Thanks for your careful reviewing, and the mentioned mistake has been corrected. Please see line 92 marked in GREEN.

After these are addressed, I feel the manuscript is ready to proceed to the publication phase.

Author’s response: Thanks again for your positive comments and every constructive suggestion. It’s our honor to be recognized by expert in this field.